# Combined analysis of microbial metagenomic and metatranscriptomic sequencing data to assess *in situ* physiological conditions in the premature infant gut

Yonatan Sher[1¤]*, Matthew R. Olm[2], Tali Raveh-Sadka[2], Christopher T. Brown[2], Ruth Sher[3], Brian Firek[4], Robyn Baker[5], Michael J. Morowitz[4,5], Jillian F. Banfield[1,2]*

**1** Department of Environmental Science, Policy, and Management, University of California, Berkeley, Berkeley, California, United States of America, **2** Department of Plant and Microbial Biology, University of California, Berkeley, Berkeley, California, United States of America, **3** Enview, Inc., San Francisco, California, United States of America, **4** Department of Surgery, University of Pittsburgh School of Medicine, Pittsburgh, Pennsylvania, United States of America, **5** Magee-Womens Hospital of UPMC, Pittsburgh, Pennsylvania, United States of America

¤ Current address: Department of Biotechnology, MIGAL—Galilee Research Institute, Kiryat Shmona, Israel
* jbanfield@berkeley.edu (JFB); sher.yoni@gmail.com (YS)

**Data Availability Statement:** RNA reads are available on the short-read archive (SRA) on NCBI, Bio-project ID PRJNA505710. Scaffolds from

## Abstract

Microbes alter their transcriptomic profiles in response to the environment. The physiological conditions experienced by a microbial community can thus be inferred using meta-transcriptomic sequencing by comparing transcription levels of specifically chosen genes. However, this analysis requires accurate reference genomes to identify the specific genes from which RNA reads originate. In addition, such an analysis should avoid biases in transcript counts related to differences in organism abundance. In this study we describe an approach to address these difficulties. Sample-specific meta-genomic assembled genomes (MAGs) were used as reference genomes to accurately identify the origin of RNA reads, and transcript ratios of genes with opposite transcription responses were compared to eliminate biases related to differences in organismal abundance, an approach hereafter named the "diametric ratio" method. We used this approach to probe the environmental conditions experienced by *Escherichia* spp. in the gut of 4 premature infants, 2 of whom developed necrotizing enterocolitis (NEC), a severe inflammatory intestinal disease. We analyzed twenty fecal samples taken from four premature infants (4–6 time points from each infant), and found significantly higher diametric ratios of genes associated with low oxygen levels in samples of infants later diagnosed with NEC than in samples without NEC. We also show this method can be used for examining other physiological conditions, such as exposure to nitric oxide and osmotic pressure. These study results should be treated with caution, due to the presence of confounding factors that might also distinguish between NEC and control infants. Nevertheless, together with benchmarking analyses, we show here that the diametric ratio approach can be applied for evaluating the physiological conditions experienced by microbes *in situ*. Results from similar studies can be further applied for designing diagnostic methods to detect NEC in its early developmental stages.

assemblies are accessible on ggkbase interface (https://ggkbase.berkeley.edu/), with a 'scaffold to bin' file available as a supplementary file that can be used to reconstruct all genome bins used in this study.

**Funding:** This research was supported by the National Institutes of Health (NIH) under award no. RAI092531A to JFB. This work was supported in part by the March of Dimes Foundation research Grant no. 5-FY10-103 (to MJM). YS work was supported by postdoctoral fellowship grant no. 2016-67012-24717 from the USDA National Institute of Food and Agriculture. The funder Enview, inc., provided support in the form of salaries for the author RS, but did not have any additional role in the study design, data collection and analysis, decision to publish, or preparation of the manuscript. The specific roles of these authors are articulated in the 'author contributions' section.

**Competing interests:** RS affiliation to Enview, inc., does not alter our adherence to PLOS ONE policies on sharing data and materials.

# Introduction

Physiological conditions within the gut are important metrics to measure when studying gut inflammatory diseases [1,2], yet are notoriously difficult to measure *in vivo*. Transcriptional profiling provides information on the pool of genes that microbial cells express, and therefore can reveal the physiological conditions experienced by these cells. Microbial transcriptional patterns have been analyzed using many methods, including reverse transcription quantitative PCR [3], microarrays [4], and in recent years, RNA sequencing [5]. Analyzing transcription patterns within microbial communities, i.e. meta-transcriptomics, is challenging because it is necessary to specifically identify the microbial species from which transcripts originate. In addition, it is also important to account for changes in the abundance of different microbial species from which transcripts originate, as changes in transcript abundance can also be related to changes in organismal abundances [6].

A relevant case in which such an approach would be useful is the study of necrotizing enterocolitis (NEC) in premature infants. The hallmark of NEC is inflammation of the small and/or large bowel that can progress rapidly to intestinal necrosis, sepsis, and death [7,8]. Because the onset of the disease is often fulminant, treatment options for severe cases are limited and often futile. Thus, the need for disease biomarkers, to enable early and accurate diagnosis, motivates ongoing research on early stages of NEC development. A recent meta-analysis of 14 DNA-based studies reported that fecal samples from preterm infants, later diagnosed with NEC, contained modest but statistically significant increased abundance of facultative anaerobes from the Proteobacteria phylum and a modest decrease in abundance of strict anaerobes [9]. As others have previously noted, an understudied approach to effectively study the bacterial response to local conditions within the infant gut is to pair taxonomic profiling with functional information [9,10]. This understudied approach can be addressed by applying the transcriptomic analysis approaches described here.

Here we established a new method of measuring transcriptomic data, named the diametric ratio, to measure physiological conditions from microbial community transcriptomic data. We used this approach to study NEC in a pilot cohort of premature infants, to infer about the transcriptional patterns associated with physiological conditions occurring before NEC is diagnosed, specifically targeting genes related to oxygen exposure.

# Methods

## Study design and sampling

This study made use of a previously analyzed dataset of metagenomic DNA sequencing of premature infant stool samples [10–13]. In this study a subset of these stool samples from four infants, two of which were diagnosed with NEC, were additionally subjected to RNA sequencing. As previously described [12], stool samples for establishing these datasets were collected after perineal stimulation [14], so that fecal samples were collected under direct vision immediately upon evacuation, to minimize changes in the transcription pattern of gut microbes outside the intestine. After collection of the samples, they were stored at -80˚C until DNA and RNA extraction. Medical information for these infants is presented in Table 1, and sampling schedule for each infant is available in Table 2. To analyze changes in transcription patterns over time and with a balanced distribution of samples, five samples from both NEC and Control infants from day of life (DOL) 10–19 were grouped over the 1st time block, and those from DOL 20–37 were grouped over the 2nd time block (Table 2).

**Table 1. Infant medical information.**

| Infant | Gestational age (Weeks) | Study | Gender | Delivery | Weight (g) | Feeding | Condition | NEC Diagnosis (DOL) |
|---|---|---|---|---|---|---|---|---|
| 64 | 28 | NIH2 | M | Vaginal | 1100 | Combination | Control | – |
| 66 | 28 | NIH2 | F | Vaginal | 1028 | Breast | Control | – |
| 69 | 26 | NIH2 | M | C-section | 637 | Combination | NEC | 32 |
| 71 | 25 | NIH2 | M | C-section | 754 | Combination | NEC | 31 |

## DNA and RNA sequencing

Procedures for DNA extraction and sequencing were previously described [13]. RNA was extracted from selected stool samples using MOBIO PowerMicrobiome RNA isolation kit. The only modification to the manufacturer's protocol was that phenol:chloroform:isoamyl alcohol was added to the glass bead tubes prior to the addition of the stool sample. RNAseq libraries were prepared with Illumina's 'TruSeq Stranded RNAseq Sample Prep kit'. Prior to library preparation, eukaryotic rRNA was removed using the 'Ribo-Zero rRNA Removal Kit (Human/Mouse/Rat)', and bacterial rRNA was removed using the 'Ribo-Zero rRNA Removal Kit (Bacteria)' from Illumina. DNA and RNA were sequenced on HiSeq2500, RNA sequencing was performed at the Roy J. Carver Biotechnology Center at the University of Illinois at Urbana-Champaign. DNA sequencing yielded an average of 35,127,007 reads per sample, while RNA sequencing of corresponding samples yielded an average of 13,786,716 reads per sample.

## Genome reconstruction

DNA reads were trimmed using Sickle (v1.33) (https://github.com/najoshi/sickle), and assembled into scaffolds with IDBA_UD (v1.1.1) [15]. Scaffolds were binned into genomes using DAS tool (v1.0), which uses a combination of established binning algorithms [16]. Reconstructed genomes from each infant were de-replicated according to 99% average nucleotide identity using dRep (v2.3.2) [17]. One genome of each identified bacterial species was manually chosen from each infant gut microbiome for downstream RNA analysis. Scaffolds from MAG's are accessible on ggkbase interface (https://ggkbase.berkeley.edu/), with a 'scaffold to bin' file available as supporting information files that can be used to reconstruct all selected genome bins used in this study (S3–S6 Files). In addition, Table A in S1 File shows accessions for the different *Escherichia* spp. genes used in this study.

**Table 2. Infant stool sampling scheme.**

| Diagnosis | infant | 1st time block | | | | | | | | 2nd time block | | | | | | | | | |
|---|---|---|---|---|---|---|---|---|---|---|---|---|---|---|---|---|---|---|---|
| | | 10 | 11 | 13 | 14 | 15 | 16 | 17 | 19 | 20 | 21 | 25 | 27 | 28 | 31 | 32 | 33 | 36 | 37 |
| **NEC** | 71 | ✓ | | | | ✓ | | | ✓ | ✓ | ✓ | ✓ | | | * | | | | |
| | 69 | | | ✓ | | | ✓ | | | | | ✓ | ✓ | | | * | | | |
| **Control** | 66 | | | | | | ✓ | ✓ | | | | | | ✓ | | ✓ | | | ✓ |
| | 64 | | ✓ | ✓ | | | ✓ | | | ✓ | | | | | | | | ✓ | |
| | | | | | | | | | | Days of life | | | | | | | | | |

✓ indicate dates stool samples were taken

* indicate NEC diagnosis

## Phylogenetic profiling

Taxonomic classification was done according to 'shared affiliation of predicted proteins' procedure, which compares each predicted protein on genome scaffolds, assembled from metagenomic sequences, to UniProt database, as described previously [11]. When more than 50% of predicted proteins on a scaffold shared the same taxonomic affiliation, which can be on any taxonomic level (from species to kingdom), this scaffold was classified according to the shared taxonomic affiliation [11].

## Gene identification and annotation

Open reading frames (ORFs) were predicted using Prodigal (v2.6.3) [18] with the option to run in metagenome mode selected. Sequences of predicted ORFs were annotated using Hidden Markov Models (HMM) [19]. Annotations of the set of genes later analyzed in transcript ratio analyses were also confirmed by aligning against UniProtKB and UNIREF100 databases.

## Calculating diametric ratios (DR)

Before analysis RNA reads were trimmed using Cutadapt [20], these RNA reads are available on the short-read archive (SRA) on NCBI, Bio-project ID PRJNA505710. To calculate Diametric Ratios (DR), RNA reads were mapped to the nucleotide sequence of all open reading frames, identified by Prodigal, in each scaffold of the de-replicated genomes sets, of each infant, using Bowtie2.

Further filtering of RNA reads, mapped to each gene sequences, was performed using the script 'mapped.py' (https://github.com/christophertbrown/bioscripts/blob/master/ctbBio/mapped.py). This script filters out any reads that map with more than one mismatch to the reference genome.

Transcript abundance was measured as coverage depth, by calculating the number of RNA reads per gene length. To calculate RNA reads coverage depth on each gene sequence we used the script 'calculate_coverage.py' (https://github.com/christophertbrown/bioscripts/blob/master/ctbBio/calculate_coverage.py). mapped.py and calculate_coverage.py are part of ctbbio version 0.45.

Gene expression levels were measured using diametric transcript ratios (*DR*), calculated according to Eq (1):q

$$DR(G, E) = \frac{\overline{RNA(\alpha_G)}}{\overline{RNA(\alpha_G)} + \overline{RNA(\beta_G)}} \qquad \begin{array}{l} \textit{where } \alpha \textit{ are genes within G that increase in response to E} \\ \textit{and } \beta \textit{ are genes within G that decrease in response to E} \end{array} \qquad (1)$$

Where α is the transcript (*RNA*) abundance of a certain gene/genes (average abundances if abundances of several genes are considered, as denoted with the overbar) in a specific genome (*G*), and β is the transcript (*RNA*) abundance of a different gene/genes in the same genome (*G*) with an opposite transcriptional response to a surrounding physiological condition (*E*).

For example, if a certain surrounding physiological condition (*E*; e.g. oxygen level) increases transcription of gene α in genome *G* than for the same physiological condition (*E*) transcription of gene β in genome *G* decreases (Table 3). Because of the opposite transcriptional responses of examined genes these transcript ratios are referred to as diametric ratios (DR). Coupled genes for DR must have a transcriptional response that is opposite to a given physiological condition (such as *norVW* vs *norR*; Table 3) or respond to opposing physiological condition (such as *ompC* vs *ompF* Table 3). Calculating transcriptional responses this way allows measuring shifts in transcriptional responses while minimizing biases associated with changes in genome abundance. As noted, transcripts of the two sets of genes (α and β)

**Table 3. Genes examined in this study and the factors controlling their transcription.**

| Coupled genes for DR | Genes | Factors controlling transcription§ | Variable in Eq (1) | Reference |
|---|---|---|---|---|
| 1) | cydAB | Micro-aerobic↑ | α | [21] |
| 1) | cyoABCD | Aerobic↑ | β | [21] |
| 2) | arcA | Anaerobic ↑ | α | [22] |
| 2) | fnr | Oxygen in-depended | β | [23] |
| 3) | nrdDG | Anaerobic and Micro-aerobic↑ | α | [24] |
| 3) | nrdAB | Aerobic↑ | β | [24] |
| 4) | norVW | Nitric oxide ↑ | α | [25] |
| 4) | norR | Nitric oxide ↓ | β | [25] |
| 5) | ompC | High osmolarity ↑ | α | [26] |
| 5) | ompF | Low osmolarity ↑ | β | [26] |

§ Arrow direction indicates whether the controlling factor up or down regulate transcription of the gene.

originate from the same genome (*G*). Thus, given that the values of α and β are composed from transcript abundance per genome multiplied by genome abundance, the DR formula factors out genome abundance and the DR are comparable between samples.

In each sample of our DR analysis we discarded genes that didn't have any RNA reads mapped to them after the stringent filtering, to avoid α or β values of 0 leading to extreme DR values of 1 or 0.

## Mapping to cytochrome oxidases from infant MAG's compared to cytochrome oxidases from KEGG

In order to compare between mapping of RNA reads to genes from assembled genomes from premature infant's gut microbiome to mapping of RNA reads to genes from available databases, *E. coli* (K-12 MG1655) cytochrome oxidases sequences were downloaded from KEGG (Kyoto Encyclopedia of Genes and Genomes) database. RNA reads were mapped to these genes and filtered as described previously for genes from assembled genomes. Diametric ratios were created for those mappings.

## Statistical analyses

Differences in transcript ratios were visualized with boxplots constructed with *R* software for statistical computing (v3.5.1), using *ggplot2* package (v3.1.1) [27].

Differences in DR of examined microbial genes were compared within time blocks between NEC and control infants. The first-time block was from 10[th] day of life till 19[th] day of life and a second time block was from 20[th] day of life till 36[th] day of life. In addition, differences in the DR of examined microbial genes were compared between all NEC and all control infants. Comparisons were done with Welch's t-test. To avoid type 1 error due to multiple comparisons, *p* values were adjusted with Bonferroni correction.

## Results

### Establishment of the diametric ratio as a quantitative metric

In this study, a new approach was examined to circumvent potential biases associated with meta-transcriptomic analysis. This approach involved accurate mapping of RNA reads to metagenomic assembled genomes (MAG's) from the same pool of samples from which RNA reads were retrieved, to be more confident to the genomic origin of the transcripts. Next,

transcript abundance ratios of genes that are known to have opposing transcriptional responses to a specific environmental exposure were calculated. To avoid biases related to changes in genome abundances, ratios were calculated only using transcripts belonging to the same genome within the same sample. This approach was designated as Diametric Ratios (DR) analysis, due to the expected opposite expression patterns of chosen genes. The approaches described here are hypothesis-driven, and are distinct from other recent approaches described for meta-transcriptomic observations, which aim to achieve a wide-spread comprehensive view of the changes in relative abundances of gene families and pathways [6,28].

To assess this approach, we analyzed 20 meta-transcriptomic datasets paired with MAG's from gut microbiomes of 4 premature infants. Two of those premature infants were eventually diagnosed with NEC.

We focused our analysis to *Escherichia* spp. as this genus was ubiquitous in almost all of the analyzed samples (as representatives of this genus occurred in most of the samples, allowing statistical analysis between NEC and control samples), while other species or genera were not adequately ubiquitous for conducting comparisons between NEC and control samples (Fig A in S1 File). More than 85% of the predicted proteins on the same scaffolds of each of *Escherichia* spp. genomes had shared affiliation with *Escherichia* genus. On the species taxonomic level more than 84% of the predicted proteins on the same scaffolds had shared affiliation with either *E. coli* or *E. vulneris* (Table B in S1 File). Relative abundances of different bacterial genera were calculated by the ratio of genome coverage to sum of coverages of all genomes in a sample (Fig A in S1 File). Coverage depth and relative abundances data of other MAG's in each sample is also available in S3–S6 Files. Each sample included about 15 MAG's (max-24, min-10), with the genus *Escherichia* occurring in most samples. It is important to note, that RNA reads from each sample were mapped to the *Escherichia* sp. genome found in the same sample, as not all of the infants had *E.coli* species. Subsequent analyses were carried out on the genus level of *Escherichia*. Therefore, calculated DRs for all *Escherichia* spp. found in the different samples were gathered and compared between NEC infants and control infants. As *Escherichia* spp. are facultative anaerobes, they can adjust their life style according to available oxygen, which was well represented in its transcription pattern [3]. Thus, by examining *Escherichia* spp. we addressed a longstanding idea that inadequate oxygen tension in the intestine, due to reduced regional blood flow (ischemia), may be a key contributor to NEC development [29,30]. *In vivo* patterns of microbial gene expression in the infant gut may be an indicator of insufficient oxygen supply to the intestine and the progression of NEC. Furthermore, diametric ratios of different sets of genes related to other physiological conditions, such as exposure to different nitric oxide (NO) and osmolarity levels, were examined as well.

We examined transcripts abundances of *cydAB* and *cyoABCD*, genes encoding cytochrome oxidases with high and low affinity to oxygen (respectively; see below '*cydAB cyoABCD* diametric ratio' section). Transcript abundances of each of these two sets of genes (either *cydAB* or *cyoABCD*, without any normalization accounting for genome abundance) was extremely variable within all-time blocks of either NEC or control infants. This high variation was exemplified through the high standard deviation compared to the average transcript abundance, yielding high coefficients of variance (between 93–174; Table 4). However, in NEC samples there was a trend showing differences between abundances of *cydAB* genes compared to *cyoABCD* genes while in Control infants no differences were found, inspiring the idea of diametric ratios. After DR calculation, much lower variation was found, yielding much lower coefficients of variance (20 and 24 for NEC and control infants, respectively; Table 4).

**Table 4. Comparison between transcript abundances and diametric ratio (DR) of *Escherichia* spp. cytochrome oxidases across all time blocks in NEC and control infants.**

| Infants | Genes/DR | Average | Standard Deviation | Coefficient of Variance | F-TEST[$] | T-TEST[$] |
|---------|----------|---------|--------------------|-----------------------|-----------|-----------|
| **NEC** | *cydAB* | 233 | 217 | 93.0 | 1.53E-12 | 0.008 |
| | *cyoABCD* | 71.7 | 125 | 174 | 2.03E-09 | 0.117 |
| | *cydAB-cyoABCD* **DR** | 0.82 | 0.17 | 20.3 | 0.226 | 2.11E-05 |
| **Control** | *cydAB* | 2.18 | 2.73 | 125 | | |
| | *cyoABCD* | 3.09 | 4.40 | 142 | | |
| | *cydAB-cyoABCD* **DR** | 0.43 | 0.10 | 23.9 | | |

[$] p-values

## *cydAB cyoABCD* diametric ratio

To examine the association between microbiome oxygen exposure and NEC development in the gut of premature infants, we first examined genes encoding for cytochrome oxidases. These protein complexes are a part of the electron transport chain that pass electrons to $O_2$ during aerobic respiration. We constructed DR from transcript abundances of bacterial cytochrome oxidase genes that have different affinities for oxygen: cytochrome bd oxidase (*cydAB*) with high affinity for oxygen and cytochrome o oxidase (*cyoABCD*) with low affinity for oxygen [31,32]. Consistent with these biochemical predictions, a previous study showed that under microaerophilic conditions there was higher expression of *cydAB* whereas under aerobic conditions expression of *cyoABCD* genes increases [21]. Our results showed that *Escherichia* spp. had higher *cydAB* to *cyoABCD* transcript ratio in the gut of NEC infants compared to control infants (Fig 1A), and these differences were significant within both time blocks ($p < 0.05$) and also across all time blocks ($p < 0.01$).

To evaluate whether mapping RNA reads to genes from sample-specific MAG's could improve results sensitivity compared to mapping to genes retrieved from KEGG databases, we examined results of *cydAB* and *cyoABCD* diametric ratios between the two mapping approaches. We found that mapping RNA reads to genes from *Escherichia* spp. from each infant microbiome MAG's had more significantly distinguishable DR between NEC and control infants then DR found for RNA reads mapped to genes retrieved from KEGG database (Fig 1A and Fig 1B, respectively). Further evaluation of differences between MAG's and *E.coli* K12 gene sequences through alignments of *cydA* genes showed variable number of mismatches (S8 File), ranging from 1 to 19 in *E.coli* species and 269 mismatches with *E.vulneris*. This signal was enhanced when filtering out reads that had more than 1 base mismatch (Fig 1A compared to Fig 1C). After filtering, mapping to KEGG database genes was even less assured as there were less data points due to filtering out of reads that were inaccurately mapped (Fig 1B compared to Fig 1D). These results highlighted the necessity for having MAG's retrieved from the same sample set as RNA reads were retrieved, as well as accurate mapping of RNA reads.

## *fnr arcA* and *nrdGD nrdAB* diametric ratios

To further strengthen our observations that *Escherichia* spp. in the gut of NEC infants were exposed to lower oxygen levels, we analyzed another set of genes that are differentially transcribed in response to oxygen levels, *fnr* and *arcA* (**F**umarate and **n**itrate **r**eductase, and **A**erobic **r**espiration **c**ontrol protein, respectively). *fnr* transcription is consistent in both aerobic and anaerobic conditions [22], and the regulatory mode is through changes in FNR protein

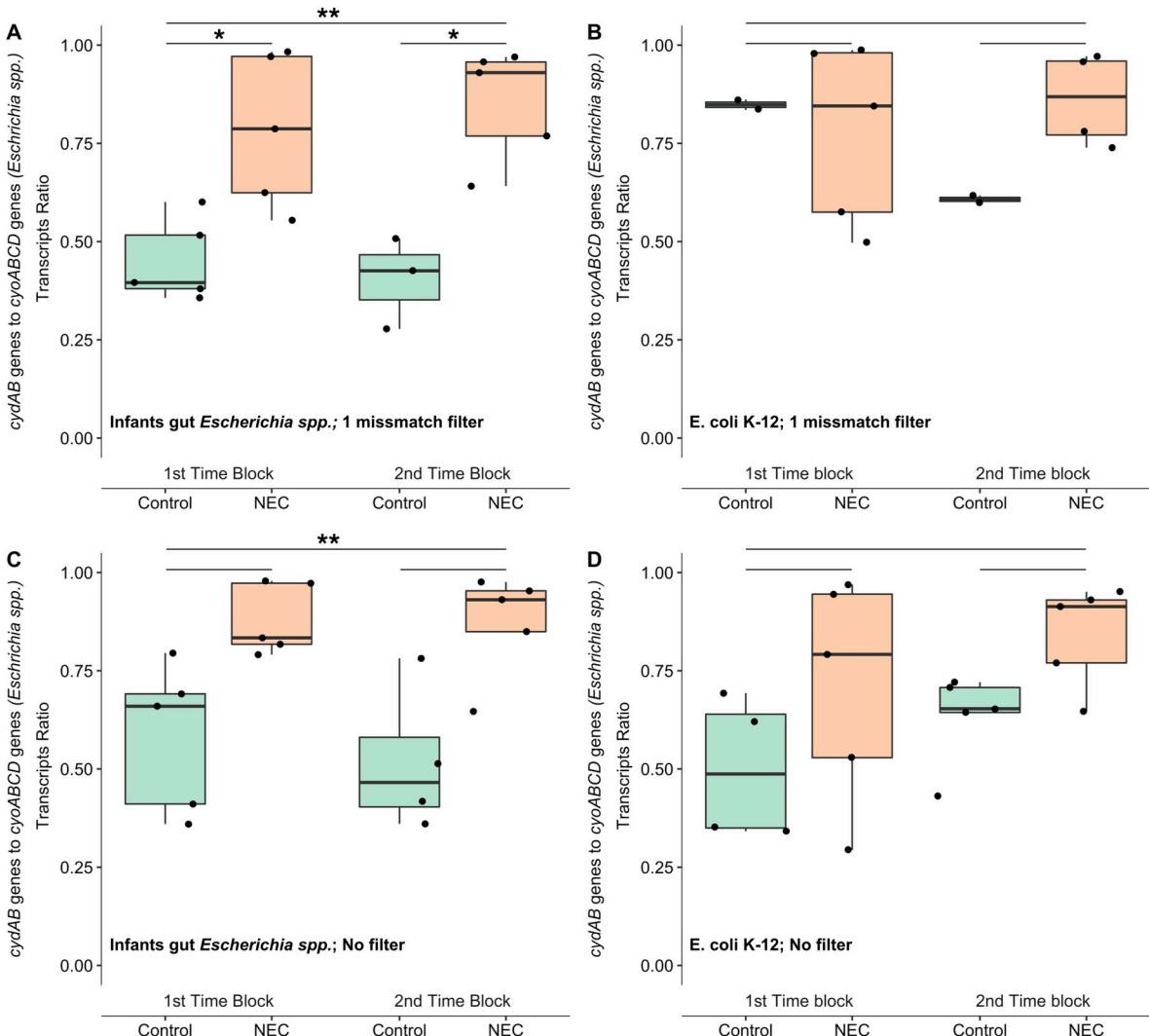

**Fig 1. Transcriptional response to oxygen by *Escherichia* spp. in the gut of NEC and control premature infants.** (A) Diametric ratios were compared between NEC and control infants in each time block (short lines above) and across all time points (longer lines). Distributions of diametric ratios were compared using Welch's t-test with Bonferroni correction. Asterisks and double asterisks (*, **) represent $p < 0.05$ and $p < 0.01$, respectively. (B) Diametric ratios of *cydAB* and *cyoABCD* transcript abundances of RNA reads mapped to gene sequences of *Escherichia* spp. genomes found in infants' gut. Filtering of reads with more than 1 miss matches was applied. (C) Diametric ratios of *cydAB* and *cyoABCD* transcript abundances of RNA reads mapped to gene sequences of *E. coli* (K-12 MG1655) downloaded from KEGG (Kyoto Encyclopedia of Genes and Genomes) database. Filtering of reads with more than 1 miss matches was applied. (D) Diametric ratios of *cydAB* and *cyoABCD* transcript abundances of RNA reads mapped to gene sequences of *Escherichia* spp. genomes found in infants' gut. No filtering of reads with miss matches was applied.

conformation at different oxygen levels [33]. However, the transcription of *arcA*, which is regulated by *fnr*, increases when low oxygen conditions prevail [22].

A third set of genes that their transcription is associated to oxygen levels, *nrd*, encodes ribonucleotide reductase, which catalyzes the enzymatic reduction of ribonucleotides to deoxyribonucleotides. *Escherichia* spp. have two sets of *nrd* genes that are differently expressed depending on prevailing oxygen levels. Under aerobic conditions transcription of *nrdAB* is upregulated, while under anaerobic condition transcription of *nrdDG* is upregulated [24].

According to the *cydAB* and *cyoABCD* diametric ratios we hypothesized that diametric ratios of *arcA* and *fnr* genes and *nrdDG* and *nrdAB* genes transcribed by *Escherichia* spp. in

the gut of NEC infants would be higher compared to control infants. Indeed, results of these diametric ratios confirmed the results of *cydAB* and *cyoABCD* diametric ratios, showing that higher ratios of both *arcA* and *fnr* genes *nrdDG* and *nrdAB* genes of *Escherichia* spp. in the gut of the NEC premature infants were significantly higher, across all time blocks ($p < 0.05$ and $p < 0.01$, respectively; Fig 2A and 2B) and specifically in the 1st time block ($p < 0.01$ and $p < 0.05$, respectively; Fig 2A and 2B). These results further suggested that *Escherichia* spp. in the gut of the NEC premature infants were exposed to lower oxygen levels.

### *ompC ompF* and *norVW norR* diametric ratios

The next set of genes we examined encode the Outer Membrane Proteins, *ompC* and *ompF*. These genes were previously shown to be transcribed under different osmotic conditions, with high abundance of *ompC* being associated with inflammatory bowel diseases [34]. High expression of *ompC*, which has small porin size, occurs during high osmotic conditions, while *ompF*, which has large porin size, occurs during low osmotic conditions [26]. These genes are reciprocally regulated by *ompR*, depending on its phosphorylation state [26]. Interestingly, significantly higher diametric ratios of *ompC* to *opmF* were found to be transcribed by *Escherichia* spp. in the gut of NEC premature infants compared to control premature infants across all time blocks ($p > 0.05$), specifically in the 1st time block ($p < 0.05$; Fig 2C).

The last set of genes we examined were the *norVW* genes, coding for NO detoxifying enzyme Nitric Oxide Reductase, and their oppositely transcribed regulating gene *norR* [25]. Nitric oxide binds to constitutively expressed NorR, which up-regulates the transcription of *norVW* and down-regulates *norR* expression [25]. Thus, to assess microbial transcriptional response to NO prior to NEC diagnosis, we measured the ratio of transcript abundances for *norVW* and *norR*. Higher diametric ratios of NO detoxifying enzymes compared to *norR* transcribed by *Escherichia* spp. were found in the guts of the control infants compared to NEC infants, in both time blocks and across time blocks ($p < 0.01$, for all cases; Fig 2D). Interestingly, the ratio of *norVW* to *norR* transcript abundances was higher at earlier compared to later time points, for both NEC and control infants (Welch's t test, $p = 0.009$). This may reflect the response of the preterm infant gut to initial microbial colonization.

## Discussion

### Using diametric ratios and RNAseq mapping to MAG's to infer physiological conditions in the gut of premature infants

Here we demonstrate that calculating diametric ratios of genes with opposite transcriptional responses to ambient physiological conditions is an approach that can be effectively used for analyzing meta-transcriptomic data (Table 4; Fig 1). In addition, we show that mapping to sample specific MAG's provides the most clear and significant signal. Results based on three sets of genes (cytochrome oxidase genes, aerobic/anaerobic regulation genes and ribonucleotide reductase genes) suggest that *Escherichia* spp. in the gut of two premature infants that developed NEC were exposed to lower oxygen levels than *Escherichia* spp. in the gut of two premature infants without NEC. Together, these data suggest that hypoxic conditions may exist in the gut prior to NEC development.

Previous animal model studies and analyses of early microbial colonization of premature infant guts showed that at early gestational age the gut milieu was more aerobic [35–38]. During NEC, however, hypoxic conditions were observed in the gut tissue of many patients [39]. Furthermore, histologic examination of removed dead intestine tissue of NEC patients demonstrated coagulation necrosis, evidence for ischemic injury [40], in either small or large

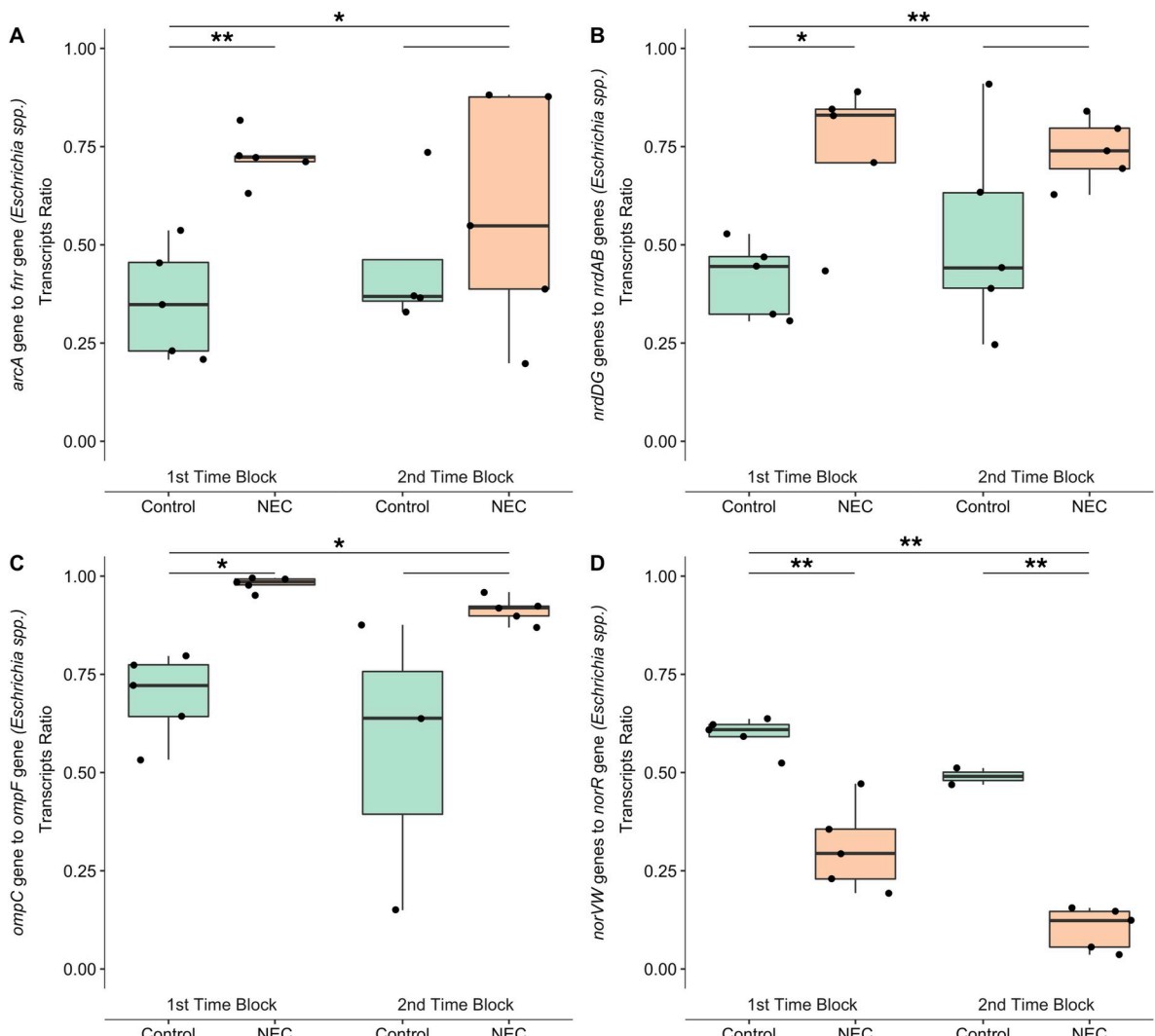

**Fig 2. Transcriptional response to oxygen, Nitric oxide and osmotic conditions by *Escherichia* spp. in the gut of NEC and control premature infants.** (A) Diametric ratios were compared between NEC and control infants in each time block (short lines above) and across all time points (longer lines). Distributions of diametric ratios were compared using Welch's t-test with Bonferroni correction. Asterisks and double asterisks (\*, \*\*) represent $p < 0.05$ and $p < 0.01$, respectively.(A) Diametric ratios of *arcA* and *fnr* transcript abundances of RNA reads mapped to sequences of *Escherichia* spp. genomes found in infants' gut. (B) Diametric ratios of *nrdDG* and *nrdAB* transcript abundances of RNA reads mapped to sequences of *Escherichia* spp. genomes found in infants' gut. (C) Diametric ratios of *ompC* and *ompF* transcript abundances of RNA reads mapped to gene sequences of *Escherichia* spp. genomes found in infants' gut. (D) Diametric ratios of *norVW* and *norV* transcript abundances of RNA reads mapped to gene sequences of *Escherichia* spp. genomes found in infants' gut.

intestine. Yet, hypoxia is highly debated as a primary controlling factor of NEC [29,41–43]. Recent theories on NEC development point to the role of gut tissue immaturity, impairing intestinal microcirculation and oxygen delivery [44], which might explain the observed transcriptional response to lower $O_2$ levels in the gut microbiomes of NEC infants (Fig 1A; Fig 2A and 2B). It should be noted also that these circumstances are distinct from those that occur in mature gut systems, where anaerobic conditions are linked to a healthy condition and aerobicity is linked to inflammation [45].

Another potential indicator for progression of inflammatory response in the infant gut is exposure of the gut microbiome to nitric oxide (NO). Increased expression of inducible nitric

oxide synthase (iNOS) by host's gut epithelial cells often occurs as a part of the inflammatory response, and recent studies have suggested that TLR4-mediated iNOS expression is a key element of NEC progression [46]. Thus, diametric ratios for nitric oxide reductase can help infer about exposure of *Escherichia* spp. to different nitric oxide levels in the gut of premature infants. Counter-intuitively to progression of an inflammatory response in the gut of NEC infants, results of this study indicate down regulation of bacterial genes for NO detoxification in the gut of NEC infants (Fig 2D), indicating lower NO levels in the gut lumen. This might be explained by low oxygen supply to the gut lumen, as suggested by results found by previous examined genes (Fig 1A; Fig 2A and 2B), reducing host epithelial cells iNOS activity to produce the antimicrobial agent NO, as these enzymes need oxygen to produce NO [47,48]. Alternatively, these results might also be explained by the report that *norVW* transcription decreases with combined oxidative and nitrosative stresses in contrast to nitrosative stress alone [49], as occurs during inflammation [50]. Inflammatory response inducing combined oxidative and nitrosative stresses also stimulates higher *cydAB* transcription (Fig 1A; [51,52]). An additional physiological condition that can be associated with inflammatory response is altered osmotic conditions [53].

Results of diametric ratios between transcripts of outer membrane protein genes indicate that *Escherichia* spp. in the gut of NEC premature infants might be exposed to high osmotic conditions (Fig 2C). Consistent with previous observation of high expression of *ompC* by *E. coli* during high osmolarity levels and increased adherence to host gut epithelial cell through the development of Crohn's disease [34]. To the best of our knowledge, little is known about gut lumen osmolarity and the development of NEC. Osmolality of feeding formula and its association with NEC development has been studied, but no clear connection was found [54].

## Confounding factors of this study dataset

It should be noted, however, that this data set contains confounding factors limiting the interpretation of *Escherichia* spp. transcriptional differences between NEC and control infants solely as a result of NEC development. First of all, the limited number of infants (two NEC and two control) examined in this study impaired our ability to draw conclusions. Low sample number also restricted our ability to define cutoffs to optimize DR analysis, as discarding more data points would have limited statistical analysis on the time block level. Secondly, other factors, such as gestational age, mode of delivery and birth weight, also differed between NEC cases and controls (Table 1). These factors were previously shown to affect the development of microbial communities [36,55], suggesting that occurrence of different physiological conditions bring about such alterations in infant gut microbiomes.

Nevertheless, transcription results shown here were opposite than expected according to some of those factors, such as gestational age. According to previous studies, younger gestational age infants might have more aerobic conditions as their microbial communities are associated with a more facultative anaerobic life style compared to older gestational age infants associated with an obligate anaerobic life style [36,38,56]. Whereas our results show that lower oxygen exposure occurred at earlier gestational age, in infants that develop NEC (Table 1; Fig 1A; Fig 2A and 2B). Although, larger sized and higher time point resolution gut microbiome meta-transcriptomic studies would be fundamental to confirm shifts in aerobicity state in the intestine of premature infants. In addition, many of the studies examining how these factors affect microbial communities were done on full term infants and data on gut microbiome of preterm infants is still very scarce. A recent paper examining gut microbiome in preterm infants showed that cesarean or vaginal birth mode did not significantly affect microbial

communities [57], unlike full-term infant where delivery mode is a major factor shaping infant gut microbiome [55,58].

Although it is hard to conclude whether observed differences were associated with NEC development or other factors, the methodological approach described here still shows a clear and significant signal distinguishing between NEC and control premature infants. A larger study is needed to verify these results and confirm that the gut microbiome in early stages of NEC development senses and responds to different physiological conditions compared to microbiomes in guts of premature infant where NEC is not developing. Further experiments can verify this approach, either by experiments where stool samples are inoculated into artificial media with varying physiological condition (e.g. oxygen levels or NO) or experiments with animal models inducing wanted physiological conditions *in-vivo*.

## Concluding remarks on examined approaches

The approach described here can add new insights into gut microbiome analysis. It is important to note that this approach relies upon prior knowledge on the transcriptional responses of different genes to physiological conditions. It is, thus, essential to do a preliminary literature survey on gene expression pathways of dominant species in examined samples to decide on genomes and genes from which to calculate diametric ratios. In addition, a significant factor affecting the accuracy of DR measurements is proper read mapping. Mapping RNA reads to sample-specific MAG's is shown here to give more significantly distinguishable signal compared to mapping to genes retrieved from an outside database (Fig 1). The mapping exercise shown here might potentially describe the extreme end of strain heterogeneity, as K-12 *E. coli* is a laboratory strain that can be different enough from gut *E. coli* to result in inquorate RNA read mapping. Choosing genomes from databases originating from gut microbiome databases might be more adequate for RNA mapping than KEGG database, as long as chosen genomes are most similar to those genomes for which RNA was sequenced. Genome databases are valuable for taxonomic identification, as done here for confident identification of MAG's in our samples. However, based on propositions put forth in this study, further research might further enforce the idea that MAG's enable more accurate mapping and promote better quantification of RNA reads than genomes from databases. It is also important to note that potentially fewer metagenomes are required per individual than meta-transcriptomes, as it was previously shown that specific strains can remain stable within an individual human host [12,59].

In conclusion, we show here how meta-transcriptomic data combined with sample-specific MAG's can be applied effectively to probe the physiological conditions that gut microbial communities experience by comparing diametric ratios. Further application of this approach can bring new insights on microbe-host interactions within the GI tract systems, and potentially help identify biomarkers for early detection gut diseases, such as NEC, onset and progression.

## Supporting information

**S1 File. Fig A.** Relative abundances of different Bacterial genera; **Table A.** Accession phrases for *Escherichia* spp. genes in on ggkbase interface; **Table B.** Percent of shared affiliation of the predicted proteins.
(DOCX)

**S2 File. Genomic relative abundances and coverage depths.**
(XLSX)

**S3 File. Scaffold to bin file, infant 64.**
(TSV)

**S4 File. Scaffold to bin file, infant 66.**
(TSV)

**S5 File. Scaffold to bin file, infant 69.**
(TSV)

**S6 File. Scaffold to bin file, infant 71.**
(TSV)

**S7 File. Diametric ratios.**
(XLSX)

**S8 File. CydA-Alignments.**
(TXT)

## Acknowledgments

We wish to thank two anonymous reviewers for their critical and constructive reviews.

## Author Contributions

**Conceptualization:** Yonatan Sher, Matthew R. Olm, Tali Raveh-Sadka, Michael J. Morowitz, Jillian F. Banfield.

**Data curation:** Yonatan Sher, Tali Raveh-Sadka, Brian Firek, Robyn Baker, Michael J. Morowitz, Jillian F. Banfield.

**Formal analysis:** Yonatan Sher, Michael J. Morowitz.

**Funding acquisition:** Yonatan Sher, Michael J. Morowitz, Jillian F. Banfield.

**Investigation:** Yonatan Sher, Jillian F. Banfield.

**Methodology:** Yonatan Sher, Matthew R. Olm, Christopher T. Brown, Ruth Sher.

**Project administration:** Michael J. Morowitz, Jillian F. Banfield.

**Resources:** Michael J. Morowitz, Jillian F. Banfield.

**Software:** Matthew R. Olm, Christopher T. Brown, Ruth Sher, Jillian F. Banfield.

**Supervision:** Michael J. Morowitz, Jillian F. Banfield.

**Writing – original draft:** Yonatan Sher, Michael J. Morowitz, Jillian F. Banfield.

**Writing – review & editing:** Yonatan Sher, Matthew R. Olm, Ruth Sher, Brian Firek, Robyn Baker, Michael J. Morowitz, Jillian F. Banfield.

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
