## [Decision Letter · Decision Letter 0]

3 Sep 2019

PONE-D-19-22108

Combined analysis of microbial metagenomic and metatranscriptomic sequencing data to assess in situ physiological conditions in the premature infant gut

PLOS ONE

Dear Dr. Sher,

Thank you for submitting your manuscript to PLOS ONE. After careful consideration, we feel that it has merit but does not fully meet PLOS ONE’s publication criteria as it currently stands. Therefore, we invite you to submit a revised version of the manuscript that addresses the points raised during the review process.

Your manuscript has been evaluated by two reviewers. As you can see in their comments, they have numerous comments concerning clarity of the study and its presentation. Hence, submitting a revision might be challenging. However, since I agree with both reviewers that the study is interesting, I give you the opportunity to submit a revised version of the manuscript. Please make sure that all concerns raised by the reviewers are well addressed in the revision. 

We would appreciate receiving your revised manuscript by 20 October 2019. To enhance the reproducibility of your results, we recommend that if applicable you deposit your laboratory protocols in protocols.io, where a protocol can be assigned its own identifier (DOI) such that it can be cited independently in the future. For instructions see: http://journals.plos.org/plosone/s/submission-guidelines#loc-laboratory-protocols

We look forward to receiving your revised manuscript.

Kind regards,

Erwin G Zoetendal, PhD

Academic Editor

PLOS ONE

Journal Requirements:

1. Thank you for including your competing interests statement; "The authors have declared that no competing interests exist."

We note that one or more of the authors are employed by a commercial company: Enview, inc.

Reviewers' comments:

Reviewer's Responses to Questions

**Comments to the Author**

1. Is the manuscript technically sound, and do the data support the conclusions?

Reviewer #1: Partly

Reviewer #2: Partly

2. Has the statistical analysis been performed appropriately and rigorously? 

Reviewer #1: No

Reviewer #2: I Don't Know

3. Have the authors made all data underlying the findings in their manuscript fully available?

Reviewer #1: No

Reviewer #2: Yes

4. Is the manuscript presented in an intelligible fashion and written in standard English?

Reviewer #1: No

Reviewer #2: Yes

5. Review Comments to the Author

Reviewer #1: Yonathan Sher et al describe a method to study physiological conditions based on bacterial gene expression, applied on feces derived from preterm infants that do or don’t develop NEC. While the approach in itself is interesting and could be applied in multiple settings, the method is not described in enough detail to be repeated by others, it is not validated under known ‘physiological conditions’ and I doubt its applicability in diagnostic setting as proposed by the authors. Since NEC most often occurs after 2 to 4 weeks of life, time is a limiting factor when using such methodology for early NEC diagnosis. See my detailed comments below.

[General]

- Many grammatical and spelling errors throughout the manuscript

- Is oxygen deprivation in the gut of preterm infants a local issue, or does it affect the complete intestine? Does it affect both the small and large intestine? What was the situation for the two included NEC infants? If affecting the small intestine in a local manner, are feces samples a good representative?

[Introduction]

Line 36: remove dash for in-vivo.

Line 64: ‘can be used to the study’ change to ‘can be used to study’.

[Methods]

Used software is without version numbers and parameters/settings and can therefore not be reproduced by others.

Lines 69-71: I encourage a more detailed description of how fecal samples were processed and stored to indicate compatibility for RNA sequencing. In addition, the manuscripts focusses on oxygen exposure, which may be challenging taking the method of fecal sample collection into consideration.

Lines 88: Change ‘which use’ to ‘which uses’.

Lines: 92-95: Does this mean these scaffolds are not yet made available? Please make them available with the next submission.

Line 99: Change ‘phyla level’ to ‘phylum level’.

Line 103: Write HMMs fully, since it is the first and only time the word is used.

Line 109-110: Please clarify what is filtered for by mapped.py. How does this complement to the mapping performed using Bowtie2?

Line 112: Use of "coverage" is ambiguous: breadth or depth? (see also https://www.danielecook.com/calculate-depth-and-breadth-of-coverage-from-a-bam-file/)

Line 116-120: This equation only takes transcript abundances into account. I wonder how this corrects for genome abundance. Did the authors considered to correct for genome abundance by using their metagenomics data? Or by other means (by correcting for total reads or total mapped reads (https://www.biostars.org/p/273537/). Also, please clarify how opposite regulating genes are identified. What does opposite regulating genes mean exactly and how can one find them for their function of interest?

[Results]

Write the section in past tense

Line 140: Change to ‘In this study new approaches were examined to…’

Line 142: Change to ‘from the same pool of samples from which RNA reads were retrieved…’

Line 143: Change ‘confidante’ to ‘confident’

Lines 140-150: Many grammatical errors, please correct this.

Line 153: The term NEC was already introduced, so the abbreviation can be used.

Lines 155-164: Move to introduction section.

Figure 1: Change the order of A,B,C and D, since now the authors first refer to figure 1D, and later on to A B and C.

Line 177: The authors describe a significant difference in cydAB to cyoABCD ratio across all time points. I wonder how statistical analysis on the separate time points has been performed, since collection of more than 1 sample on each time point was rare, as shown in table 2, and statistical analysis can therefore not be performed. Maybe the authors meant to say something different, please clarify.

Line 184: Authors refer to figure 1CD, while to understand the differences between using KEGG database of infant MAGs, one should compare A with C.

Lines 182-184: Although not significant, the same trend can be observed when using genes from the KEGG database. Could this be a matter of small sample size?

Lines 187-188: It could also be a matter of picking the right E. coli strain from the database. Why was E. coli K12 selected? This is a laboratory strain and may not reflect functional properties (and therefore genes) that are needed to thrive in the human gut. A better option may be to use public available whole genomes from clinical isolates from the NICU.

Line 193: Add full name of the genes

Line 194: Change FNR to fnr

Line 202: Change to ‘hypothesized’.

[Discussion]

Line 298-299: Did the authors consider to mimic low and high oxygen exposure in an in vitro setting or animal model to confirm their method? Would rather do that to begin with than studying a bigger group of infants. Currently, the method is not validated, but directly applied to clinical samples during which too many aspects are based on assumptions.

Reviewer #2: This paper describes an interesting approach to make more (or better) use of meta-transcriptomics data. Even though a priori knowledge is needed, thus limiting it to exploration of already established pathways and transcriptional relationships.

This idea to make use of the metagenomic derived MAG's poses an interesting approach. However, the described methodology is, in the current manuscript text, not always clear enough to follow.

General remarks:

- The tone of the abstract is rather strong. The main text is better nuanced, especially due to the section on the limitations (i.e. the Confounding factors section that describes the sample size, subject "baseline" differences, etc). Strongly suggest to tone down the abstract to prevent overselling.

- The definition and formula of the diametric ratio remain a bit vague. Though it is understandably depending on each gene, it would help to literally state the coupled genes (Table 3 does not provide direct linking). Furthermore, actual descriptive statistics on the transcript abundances of the used examples would greatly help following the method (so the A's and B's of each of the example gene sets, as the boxplots only show the final outcome ratio). Also, any given external condition/stimulus would likely affect several genes, what to do with other genes that are known to be affected by oxygen or infllammation status?

- Although reasoning to focus on E.coli is understandable, it does make the reader wonder what happened to all the other MAGs. How many were there per sample?

- In line with the previous point how well are the MAGs assigned to a taxonomic unit? How confident is the assignment to E.coli? In the Materials and Methods section the phylogenetic assignment appears to been done at the very high phylum level. How deep were the samples sequenced (both transcriptome and metagenome)? Especially since the rather short read HiSeq platform was used, a reasonable depth would be needed. These technical details, that are needed to assess the quality of the reported analysis, are missing or should be reported directly into the main text.

- Though conceptually it could make sense, the conclusion that mapping RNA reads to sample-specific MAGs is more significantly distinguishable compared to mapping to genes retrieved from an outside database is hard to follow in the text. From the results it is not particularly clear if the mentioned E.coli focus means only one (or more) found MAG(s) that was/were assigned to E.coli was/were used or not. Assuming that this is a (set of) MAG(s) identified in these samples, then the comparison appears to be made against KEGG references genes, but these only originate from one strain (K-12 MG1655)? Is the MAG assignment considered accurate to strain level (see earlier point)? If not, then why not compare to various, if not all, E.coli genes in the online repositories? Especially for E.coli there is a huge genetic variation between all the known strains. Comparing to a single set of genes can therefore not prove this is a better method.

Text is mostly well readable though some grammar checks may bring some improvements - some specific examples:

- line 196 "higher when there are more anaerobic conditions [24].": Did you mean "higher when the condition is more anerobic [24]". Perhaps it is even better to state "less/lower oxigen".

- line 236 'provide"  provides ?

6. PLOS authors have the option to publish the peer review history of their article (what does this mean?). If published, this will include your full peer review and any attached files.

Reviewer #1: No

Reviewer #2: No

---

## [Author Response · Author response to Decision Letter 0]

18 Oct 2019

We have addressed each concern raised by reviewers, in the 'response to reviewers letter', specifically addressing comments concerning clarity of the study and its presentation.

---

## [Decision Letter · Decision Letter 1]

6 Nov 2019

PONE-D-19-22108R1

Combined analysis of microbial metagenomic and metatranscriptomic sequencing data to assess in situ physiological conditions in the premature infant gut

PLOS ONE

Dear Dr. Sher,

Thank you for submitting your manuscript to PLOS ONE. After careful consideration, we feel that it has merit but does not fully meet PLOS ONE’s publication criteria as it currently stands. Therefore, we invite you to submit a revised version of the manuscript that addresses the points raised during the review process.

Your manuscript was evaluated by the same reviewers. Although they noticed some improvements, both of them were not convinced about the validity of the approach used, to which I agree. This is a major concern with the study and I was temped to reject the manuscript for publication in PLOS ONE. However, I decided a major revision as reviewer 2 pointed out that the approach could maybe be useful for others. Hence, I give your an opportunity to submit a revision. I recommend you to address all comments raised by the reviewers appropriately in a revised version. Make sure you convince us of the validity of the approach used or alternatively discuss in detail the drawbacks of it and tune down the conclusions.

We would appreciate receiving your revised manuscript by 10 December 2019. To enhance the reproducibility of your results, we recommend that if applicable you deposit your laboratory protocols in protocols.io, where a protocol can be assigned its own identifier (DOI) such that it can be cited independently in the future. For instructions see: http://journals.plos.org/plosone/s/submission-guidelines#loc-laboratory-protocols

We look forward to receiving your revised manuscript.

Kind regards,

Erwin G Zoetendal, PhD

Academic Editor

PLOS ONE

Reviewers' comments:

Reviewer's Responses to Questions

**Comments to the Author**

1. If the authors have adequately addressed your comments raised in a previous round of review and you feel that this manuscript is now acceptable for publication, you may indicate that here to bypass the “Comments to the Author” section, enter your conflict of interest statement in the “Confidential to Editor” section, and submit your "Accept" recommendation.

Reviewer #1: (No Response)

Reviewer #2: All comments have been addressed

2. Is the manuscript technically sound, and do the data support the conclusions?

Reviewer #1: Partly

Reviewer #2: Yes

3. Has the statistical analysis been performed appropriately and rigorously? 

Reviewer #1: Yes

Reviewer #2: N/A

4. Have the authors made all data underlying the findings in their manuscript fully available?

Reviewer #1: No

Reviewer #2: Yes

5. Is the manuscript presented in an intelligible fashion and written in standard English?

Reviewer #1: Yes

Reviewer #2: Yes

6. Review Comments to the Author

Reviewer #1: Yonathan Sher et al made changes to the manuscript accordingly. I, however, remain hesitant about the validity of the presented approach and drawn conclusions, because of unclarities on various levels (sample type, data processing, DR criteria, data interpretation and no validation). See detailed comments below.

[General]

Is a minimum coverage depth required for the DR to be ‘accurate’? If so, what is suggested? If not, wouldn’t that be required?

[Abstract]

Line 21-22: Organism abundance is actually taken out of the equation. Authors don’t ‘correct’ for organism abundance, because the DR is not affected by organism abundance. I suggest to rephrase this to make it more clear to the readers.

[Introduction]

Line 40-41: Suggestion to change to “Physiological conditions within the gut are important metrics to measure when studying gut inflammatory diseases [1,2], yet are notoriously difficult to measure in vivo.

Line 66-67: Based on the analysis described herein, the conclusion that the method is widely applicable cannot be drawn.

[Methods]

Line 73: Suggestion to say ‘As previously described…’ instead of ‘as noted previously…’

Line 78-79: Clarify on which basis the time blocks were stratified. Sample number-driven (so each block has 5 samples) or ‘biology’-driven, or other?

Table 1: Did any of these infants receive breathing support? Continuous positive airway pressure can result in an increased oxygen supply to the GI tract.

Line 110: This is just a link to the website, how can I find these particular scaffolds? Please add a accession number or something similar.

Line 102: In this section, it is unclear if it is about how RNA OR DNA sequences are processed. I suppose genome reconstruction is based on metagenomics data?, but in the first sentence (Line 103) RNA is mentioned as well, as well as in the later part of this section (Lines 108-111).

Line 113: Same here, I assume metagenomics data is used for phylogenetic profiling, but it is important to mention it somewhere nevertheless.

Line 125: Wouldn’t you expect to obtain more MAG’s from feces? I understand these are samples from preterms in early life, and therefore the microbiota is still minimal and developing. However, looking at the relative abundance profiles I would expect to identify more species than 15. Is it a methodological issue?

Line 125: Could the fact that most MAGs were Escherichia be a result of availability (redundancy) in the database?

Line 133: Change to “To calculate Diametric Ratios (DR), RNA reads were…”

Table 3: The table shows different ‘relationships’ between the coupled genes for DR. Some are opposing in a specific condition (e.g. number 4 in response to NO), while some conditions are opposing which results in transcription of different genes (e.g. number 5). Please clarify on which terms genes are considered opposite transcriptional responses. With other words, which criteria need to be fulfilled for genes to be identified as ‘opposite transcriptional responses’. For example, why are arcA and cyoABCD no opposing genes, as one goes up in anaerobic conditions, and the other goes up in aerobic conditions (which is a similar difference in condition as number 5 for osmolarity).

Line 173-174: Please add the version of R and the ggplot2 package.

Line 175-178: I remain unsure about which comparisons are made. Control versus NEC at time block one; Control versus NEC at time block two; and control versus NEC of all samples? I am confused because the authors write “within time block”, which I interpret as comparing samples within a time block with each other, instead of comparing all samples in a time block between control and NEC.

[Results]

Line 181: ‘a new approach was examined to..’

Line 182-184: How can this be an approach by itself? An approach to achieve what exactly? Which potential bias are circumvented by this? In the introduction, the authors focus on one new method: the diametric ratio. To my opinion, mapping of RNA reads to MAG’s is just part of the process to calculate the DR. However, two methods for mapping are examined: mapping to MAGs and mapping to the genome of E.coli K-12 from database. I suggest to focus on the DR approach and the comparison of the two mapping methods.

193: ‘to assess this approach’

Line 196: When is a genus considered adequately ubiquitous?

Line 203: The fecal microbiota does not represent the microbiota lining the mucosal layer, which may be more affected by reduced regional blood flow than luminal microbiota. In addition, the gut is not oxygen free in early life, which is important to take into consideration, particularly for the early timepoint samples (~first month, probably longer in preterm infants).

Line 211-212: ‘Transcript abundances of each of these two sets of genes (either cydAB or

212 cyoABCD, without any normalization accounting for genome abundance) was extremely variable’. Variable between what? Between samples, between NEC/control?

Line 214: Change to (between 93-174) – 93 instead of 92.

Table 4: Please add the DR for all gene sets as supplementary data.

Table 4: Average coverage of the genes in control infants was only 2-3. Getting back to my earlier comment: “Is a minimum coverage depth required for the DR to be ‘accurate’?”

Line 233-236: Escherichia spp were of much lower abundance in the control than NEC infants (Fig S1). May it be that Escherichia of low abundance express a different set of genes than dominant Escherichia, simply as a result of their abundance? Overall metabolic activity may, independent of oxygen levels, already be different. It looks like the gut environment of preterms developing NEC favors conditions for Escherichia growth and replication. Could a more competitive environment for Escherichia (like in control infants) alter expression of cydAB cyoABCD genes, since other activities may be of more importance for survival? Can the authors check for activity status of Escherichia, maybe by checking the expression of housekeeping genes, or by RNA coverage over the entire genome? Long story short: How sure are the authors that the difference in DR represents differences in oxygen levels at the epithelial lining of the gut, and not overall deprived activity in case of low abundance?

Line 258-260: Since this must be a result of RNA mapping efficiency, can the authors give an impression of the difference in gene sequence (of the specific genes of interest) between the MAGs and E coli K12?

[Discussion]

Line 330-331: On which criteria is the conclusion of ‘effectively used’ drawn? When is a method effective?

Line 412: affecting

Line 417-419: Would it be appropriate to screen the databases for Escherichia spp that result in the highest transcript abundance, and use this/these for analysis? In addition, would it be appropriate if the ‘best pick’ from the database is difference for each sample?

Line 419-422: Rebuild sentence.

Line 429: Suggestion to remove “and potentially help identify biomarkers for early detection of NEC progression in premature infants.”

Reviewer #2: The authors have greatly increased readability/transparency of the analysis. Whether or not this approach is practically suitable in preterm NEC setting is up to debate, but the approach to make use of metatranscriptomics in the described method is interesing and now much clearer for other parties to repeat.

Weaknesses and assumptions are now much more transparent.

Minor comment(s):

- Lines 107-108 "One genome of each 107 identified bacterial species was manually chosen from each infant gut microbiome for downstream RNA analysis"  I understand the manual choice here, but please clarify where this information (which bacterial species chosen for which smaples) can be found. Online? SI Table 1?

- In the Materials and Methods sections some "results" are presented among the methodology (such as the sentences in line 118-121; line 124-125). These would fit better in the Result section.

7. PLOS authors have the option to publish the peer review history of their article (what does this mean?). If published, this will include your full peer review and any attached files.

Reviewer #1: No

Reviewer #2: No

---

## [Author Response · Author response to Decision Letter 1]

10 Dec 2019

We have fully responded to of the reviewer and editor comments, as addressed in response letter.

---

## [Decision Letter · Decision Letter 2]

8 Jan 2020

PONE-D-19-22108R2

Combined analysis of microbial metagenomic and metatranscriptomic sequencing data to assess in situ physiological conditions in the premature infant gut

PLOS ONE

Dear Dr. Sher,

Thank you for submitting your manuscript to PLOS ONE. After careful consideration, we feel that it has merit but does not fully meet PLOS ONE’s publication criteria as it currently stands. Therefore, we invite you to submit a revised version of the manuscript that addresses the points raised during the review process.

The reviewers of the previous versions indicated the manuscript was majorly improved, but still had a couple of minor issues that need to be addressed. Please address these.

We would appreciate receiving your revised manuscript by 20 February 2020. To enhance the reproducibility of your results, we recommend that if applicable you deposit your laboratory protocols in protocols.io, where a protocol can be assigned its own identifier (DOI) such that it can be cited independently in the future. For instructions see: http://journals.plos.org/plosone/s/submission-guidelines#loc-laboratory-protocols

We look forward to receiving your revised manuscript.

Kind regards,

Erwin G Zoetendal, PhD

Academic Editor

PLOS ONE

Reviewers' comments:

Reviewer's Responses to Questions

**Comments to the Author**

1. If the authors have adequately addressed your comments raised in a previous round of review and you feel that this manuscript is now acceptable for publication, you may indicate that here to bypass the “Comments to the Author” section, enter your conflict of interest statement in the “Confidential to Editor” section, and submit your "Accept" recommendation.

Reviewer #1: (No Response)

Reviewer #2: All comments have been addressed

2. Is the manuscript technically sound, and do the data support the conclusions?

Reviewer #1: Partly

Reviewer #2: Yes

3. Has the statistical analysis been performed appropriately and rigorously? 

Reviewer #1: Yes

Reviewer #2: Yes

4. Have the authors made all data underlying the findings in their manuscript fully available?

Reviewer #1: Yes

Reviewer #2: Yes

5. Is the manuscript presented in an intelligible fashion and written in standard English?

Reviewer #1: Yes

Reviewer #2: Yes

6. Review Comments to the Author

Reviewer #1: Yonathan Sher et al made changes to the manuscript accordingly, including clarification of applied data processing methods and DR criteria. I remain hesitant about whether the analysis can be repeated by another party (due to the arbitrary nature of the approach) and about its applicability in clinical setting. However, the proposed method may inspire other researchers to think in this direction, and to be aware of potential biases, when analysing their -omics data.

Minor comments:

- Check grammar and spelling once more, carefully, throughout. E.g. lines 197-204: Mixed use of present and past tense (was examined, involves, were retrieved etc), same accounts for lines 214-218 (focused, was, occur, were not).

- Based on the analysis described herein, the conclusion that the method is widely applicable

cannot be drawn (the sentence stating this has already been removed in the introduction section, but not yet in the abstract).

- I suggest, for the conclusion, to focus on the approach (like lines 443-446), and not so much on the preterms/NEC. E.g. In conclusion, we show here how meta-transcriptomic data combined with sample-specific MAG’s can be applied effectively to probe the physiological conditions that gut microbial communities experience by comparing diametric ratios. Further application of this approach can bring new insights on microbe-host interactions within the GI tract systems, and potentially help identify biomarkers for disease onset/progression.

Reviewer #2: The Manuscript has been substantially improved. Validation of the approach remains a weaker point but this is much more addressed and the manuscript invites others to further explore and validate.

Some minor items to consider:

DR definition:

- The definition is based on the "opposite transcriptional response" of genes to a given environmental condition. Should this be strictly kept at transcription? Or could the "opposite" term also be "different"? Because the fnr/arcA example describes that arcA react to the condition, but the fnr transcription is (expected to be) constant so fnr transcription does not rely on the environmental condition (it's translated protein FNR does react to the condition). All other gene sets are obviously of a much more "opposite" nature.

Methodology - suggestion to enhance repeatability of procedure by others

- in the Results (line 208-210) it is stated that RNA reads were mapped on the MAGs per sample, but later this was combined per genus level. To ensure others can repeat your approach you should state how this "combining" was done. Are mapped results/data further "aggregated" based on the gene name/annotation of the different Escherichia species/MAGs?

Perhaps this is evident from the SI files, but it would make it conceptually easier to understand for a larger audience.

Suggestion to strengthen discussion in "Confounding factors" section

- the paragraph with lines 402-412 states that the presented findings are opposite to published findings related to gestational age. Though I agree that this intuitively might seem to be the case: the figures of the DRs do not seem to show a time development in either Control or NEC group, except for norVW/norV DR but this indeed seems to go in the opposite direction. However, as the breathing support is not (properly) recorded and this might have an impact. Moreover, DR findings are "relative" and no true "later gestational age / confirmed anaerobic" infant fecal sample is analyzed and such a validation could give more insight.

Some minor language items I noticed:

- line 40 "in vivo" should be in italics

- I believe "spp." should not be in italics (many times throughout the text and figure legends)

- suggestion to add the word "to" (or "with") in line 405 after "compared"

- line 407: "occur" should be "occurs"?

7. PLOS authors have the option to publish the peer review history of their article (what does this mean?). If published, this will include your full peer review and any attached files.

Reviewer #1: No

Reviewer #2: No

---

## [Author Response · Author response to Decision Letter 2]

21 Jan 2020

Thank you for indicating the major improvements of the manuscript.

We fully addressed all the minor issues pointed by the reviewers.

---

## [Editor Report · Decision Letter 3]

10 Feb 2020

Combined analysis of microbial metagenomic and metatranscriptomic sequencing data to assess in situ physiological conditions in the premature infant gut

PONE-D-19-22108R3

Dear Dr. Sher,

We are pleased to inform you that your manuscript has been judged scientifically suitable for publication and will be formally accepted for publication once it complies with all outstanding technical requirements.

I saw that (an)aerobic (for microbes) and (an)oxic (for conditions) may have been mixed up occassionally, such as for example in line 359. Make sure these terms are correctly used when checking the proof.

With kind regards,

Erwin G Zoetendal, PhD

Academic Editor

PLOS ONE
---

## [Editor Report · Acceptance letter]

21 Feb 2020

PONE-D-19-22108R3 

Combined analysis of microbial metagenomic and metatranscriptomic sequencing data to assess *in situ* physiological conditions in the premature infant gut 

Dear Dr. Sher:

I am pleased to inform you that your manuscript has been deemed suitable for publication in PLOS ONE. Congratulations! Your manuscript is now with our production department. 

With kind regards,

on behalf of

Dr. Erwin G Zoetendal 

Academic Editor

PLOS ONE